# Speeding up Smartphone-Based Dew Computing: In Vivo Experiments Setup Via an Evolutionary Algorithm

**DOI:** 10.3390/s23031388

**Published:** 2023-01-26

**Authors:** Virginia Yannibelli, Matías Hirsch, Juan Toloza, Tim A. Majchrzak, Alejandro Zunino, Cristian Mateos

**Affiliations:** 1ISISTAN (UNICEN-CONICET), Tandil 7000, Argentina; 2Department of Information Systems, University of Agder (UiA), 4604 Kristiansand, Norway

**Keywords:** dew computing, smartphones, profiling, benchmarking, Motrol, evolutionary computing

## Abstract

Dew computing aims to minimize the dependency on remote clouds by exploiting nearby nodes for solving non-trivial computational tasks, e.g., AI inferences. Nowadays, smartphones are good candidates for computing nodes; hence, smartphone clusters have been proposed to accomplish this task and load balancing is frequently a subject of research. Using the same real—i.e., in vivo—testbeds to evaluate different load balancing strategies based on energy utilization is challenging and time consuming. In principle, test repetition requires a platform to control battery charging periods between repetitions. Our Motrol hard-soft device has such a capability; however, it lacks a mechanism to assure and reduce the time in which all smartphone batteries reach the level required by the next test. We propose an evolutionary algorithm to execute smartphone battery (dis)charging plans to minimize test preparation time. Charging plans proposed by the algorithm include charging at different speeds, which is achieved by charging at maximum speed while exercising energy hungry components (the CPU and screen). To evaluate the algorithm, we use various charging/discharging battery traces of real smartphones and we compare the time-taken for our method to collectively prepare a set of smartphones versus that of individually (dis)charging all smartphones at maximum speed.

## 1. Introduction

Dew computing is an emerging paradigm that promotes the utilization of on-premise edge computing resources to execute different kinds of user applications, with the aim of reducing the dependency on remote computing resources provided by the cloud [13] and fog environments [14], and thus improving user experience due to reduced network latency [1]. 

In Dew environments, smartphones are considered as valuable on-premise computing resources [2]. This is mainly because of the capabilities of the newest smartphone models in the market in terms of computing power, battery capacity, and energy management. However, to build knowledge on the collective capabilities of these nodes in such environments, a platform that facilitates the execution and reproduction of battery-driven live tests, which are focused on studying the impact on battery usage of scavenging user’s smartphone resources is essential.

In this respect, one of the most recently proposed platforms to study smartphone capabilities in Dew environments is Motrol [3,7]. This platform offers the possibility to execute benchmarks and battery profiling tests on a set of attached smartphones, whose energy supply is controlled via a REST API. The platform is supported on Android smartphones due to the popularity of this mobile OS [15], with over 70% of the market share as of August 2022; nevertheless, the concepts presented in this paper and the broader research line are mobile OS agnostic.

Benchmarks aim to measure individual capabilities of smartphones when performing certain tasks, such as determining MFLOPs through the Linpack benchmark. Battery profiling records timestamped battery level changes registered in the context of using a combination of certain components, for example, maintaining CPU usage at 100% while the screen is on. Battery level change events can be read through the API exposed by Android [10] or iOS [11]. Collecting battery profiles of the same smartphone by varying components combined usage while combining portions of these profiles in a unique timeline allow researchers to model battery behavior based on real world data [8].

There is another type of test, called a synchronized test, which aims to study collective capabilities of smartphones in executing a compute-intensive scenario, for example, a set of distributable workloads from a specialized AI application such as object recognition from images using neural networks [9]. With this type of test, it is relevant to study different load balancing algorithms to distribute such workloads, with a common base of comparison given by measuring energy utilization disparity among smartphones performed by each algorithm. In this kind of test, the experimentation steps are as follows:(1)Define a workload;(2)Define the initial conditions for the testbed;(3)For each load balancing algorithm within the set of load balancing algorithms under evaluation:
(a)Assure initial conditions for the testbed;(b)Run the workload on the testbed using the current load balancing algorithm;(c)Collect results for further analysis.

Since real–i.e., in vivo–testbeds with battery-driven devices are involved, recreating experimental conditions, i.e., performing step 3a, is challenging. In order to make workload assignment comparisons as fair as possible, device battery levels should be restored from test to test to the value configured for the experimental scenario. The latter might require that Device A has x%, Device B has y% and Device C has z% battery levels when workload assignment of a particular load balancing algorithm starts. Thus, assuring these battery levels upon starting the evaluation of each load balancing algorithm is necessary for the sake of fair comparison.

The contributions of this work regarding the state-of-the-art methodologies to run battery-driven tests on smartphone cluster testbeds are the following:–Inclusion of a multi-device battery preparation stage as an optimization problem that uses time-related battery (dis)charging events as input;–Proposal of an evolutionary algorithm to automate and minimize the battery preparation time, evaluated using real smartphone battery traces and several combinations of smartphone cluster sizes and target battery levels;–Publicly available evolutionary algorithm and simulation engine code, experiment configuration and battery traces for reuse and modification. 

The remainder of the paper is organized as follows. In Section 2, we present an overview of the Motrol platform, designed to study smartphone capabilities in Dew environments. In Section 3, we present our novel software component for Motrol, which aims to prepare smartphones for synchronized tests using evolutionary computing. In Section 4, we present the experiments carried out to evaluate this novel component. Section 5 presents related work. Finally, in Section 6, we present the conclusions of this work and future works.

## 2. The Motrol Platform: Background on the Architecture and Basic Concepts

The Motrol platform is aimed to evaluate individual and collective capabilities of smartphones. To achieve this, this platform utilizes a backend server and an Android application named Normapp [24]. The platform allows users to attach up to four smartphones and dynamically control their energy supply via custom hardware [7] considering three energy states (no energy supply, AC power, and USB charging). We describe below the characteristics of the backend server and Normapp, and the collaboration between them to reach the aim of this platform. Moreover, we describe where on this platform the new software component proposed in this paper is located, and how the component interacts with the existing ones.

Motrol is an in-lab experimental platform that follows the client-server architectural style. In this sense, a backend server that runs on an Ri4 [12] receives jobs to be executed on the attached smartphones. The clients are the smartphones that, while rendering service to the platform, listen for jobs to be executed and send results back to the server. Currently, the platform supports two kinds of jobs: benchmarks/profiling and synchronized tests.

The main purpose of benchmarking/profiling jobs is to individually characterize smartphone capabilities [4]. Feeding the platform with a set of benchmarking/profiling jobs means that all the attached smartphones process the entire set in parallel. An example of a benchmark job is the Linpack benchmark which measures the floating-point operations a device can achieve per second. 

A profiling job, specifically a (dis)charging battery profile, captures the battery behavior of a device under certain resource usage conditions. For example, a profile might provide time-related information of battery discharging events under a configured CPU usage, e.g., 100%, and screen state, e.g., screen on, during the time the battery level starts at 100% until it reaches 1%. The output of such a profiling job is a battery trace which can then be used to model battery behavior for simulation purposes [8].

Synchronized tests, in contrast, aim to evaluate the aggregated computational capabilities of smartphone groups; hence, they are useful for evaluating load balancing algorithms [5]. This in turn is relevant, for instance, to study how different algorithms impact the smartphone’s battery usage while processing the input workload. To run this kind of test, it is required to first define a *scenario*, which consists of selecting a starting battery level (not necessarily the same) for all smartphones participating in the test and a workload. Figure 1 shows three scenario descriptor files (associated with synchronized tests) in JSON format, which have been simplified to highlight relevant parts. These parts are the participating smartphones with the desired start battery levels—which along with the configured workload, remain the same in our example for the three files—and load balancing algorithms, which vary from test to test.

Figure 2 depicts an overview of all stages involved in the execution of synchronized tests. As said, the first stage requires human intervention and involves writing at least one synchronized test specification to indicate the platform relevant runtime entities, including participating devices (smartphones models)—each of them configured with a start battery level—, a workload represented by a series of items (images) that a Java module implemented as part of Normapp processes to produce a result (e.g., object recognition associated with an specific application), and a specific load balancing algorithm to distribute the workload among participant devices. Since this stage mainly involves writing test specifications in the JSON format, there are no steps in the figure that are associated with this stage.

The second stage is semi-automatic because it requires user intervention to attach and register devices to Motrol sockets. This stage is where platform setup is performed. For a detailed explanation of this stage, please refer to [7], where the involved steps are illustrated in a video.

The last step is synchronized test plan execution, which aims to be a smart, automatic stage with the proposed evolutionary algorithm introduced in this work. A detailed look into this stage reveals three steps that are performed upon the initiation of a new synchronized test. The first step is the one targeted by the proposed solution and comprises device battery preparation. This is facilitated by a software/hardware platform [4]. However, doing this automatically while reducing preparation time so as to complete a set of synchronized tests as early as possible is not addressed by the platform. This preparation time might account for a considerable time per test, considering that while some smartphones might quickly reach the target level, others might be slower in doing so, which in turn jeopardizes the preparation of the former. It is worth mentioning that battery level preparation does not necessarily mean that all devices participating in a test should reach the same battery level. Instead, within a real Dew computing scenario, devices might collaborate with task computation while joining the group with a different battery level. The goal of this step is then to assure reproducibility of certain battery level configurations for all devices when the same scenario is exercised for different load balancing algorithms; thus, a common ground for comparison is established.

The test execution and results post processing steps are shown to complete the description of the synchronized test plan execution stage. These steps do not affect the main functioning of the proposed evolutionary algorithm.

### The Battery Preparation Problem for Running Synchronized Tests

Motrol provides a basic functionality to individually charge or discharge, i.e., *prepare*, a device battery to run a synchronized test. However, preparing devices for running a synchronized test requires a component with a global view of the preparation progress of the whole set of devices. Once a device reaches its configured battery level, e.g., x%, the platform marks it as ready for start the synchronized test, and only when all devices participating in a test are marked as ready does the test effectively start. However, by following this device preparation logic, it is not hard to fall into situations where a device (marked as ready) waits for other devices to reach their configured battery level, causing its own battery level to deviate from the required one, forcing another wait for a preparation state, the complexity of which varies with the number of devices participating in a test.

A way of avoiding this situation is by applying sequences of charge/discharge actions on the devices in the set so that all of them reach the configured battery levels at similar points in time. However, determining these sequences of actions, while keeping the time-taken for this to a minimum for the set of devices, leads to a complex combinatorial optimization problem. In this paper, we present a novel software component for Motrol that tackles this problem, which automatically prepares a set of devices to start a synchronized test. This component uses a specially designed evolutionary algorithm that has a global view of the smartphone preparation progress.

## 3. Evolutionary-based Preparation of Smartphones for Synchronized Test Plans

We developed a novel software component for Motrol, which is aimed to automatically prepare a given set of smartphones for executing a synchronized test. This component considers a usual pre-condition of synchronized test plans. This pre-condition imposes pre-defined start battery levels for the smartphones in the set. Then, the component utilizes a specially designed evolutionary algorithm that produces a sequence of charging/discharging actions to prepare the whole set of smartphones so that this pre-condition is satisfied. Specifically, the algorithm is aimed at reaching the configured start battery levels in the minimum possible amount of time for the set of smartphones. We detail below the input to this component. Then, we describe in detail the general behavior in terms of evolutionary computing used by this component.

### 3.1. Input Data of the Component and Pre-Condition Considered

The component receives as input data a given set of *m* smartphones attached to Motrol. Each one of the smartphones in the set has a known initial current battery level, where this level is represented as an integer in the range [0,100]%. In addition, each smartphone has associated timestamped battery charge/discharge profiles. Battery profiles represent realistic time-related information of a smartphones charging/discharging behavior, i.e., they allow the evolutionary algorithm to know the time (in milliseconds) required to charge/discharge the battery of the smartphone from a given level to another given level.

We firstly collected discharging and charging profiles in a laboratory context by following the procedure documented in [8]. Capturing battery profiles is a time-consuming task, especially when considering the energy consumption impact of all hardware components and its different operational modes, e.g., DVFS (dynamic voltage frequency scaling), for CPUs. Then, for practical reasons but without losing generality, we obtained profiles targeting certain combinations of discrete CPU usages <0, 30, 50, 75, 100> (in %) and screen states (on/off), resulting in different energy consumption rates. Figure 3 and Figure 4 show a timeline representation of all charging and discharging battery profiles for the smartphones considered in the experiment. Combinations of CPU usage and screen state were not deliberately selected but aim to represent the common mobile usage patterns. For instance, a profile with 0% CPU usage and an off screen represents an idle smartphone; a profile with 30% CPU usage and an on screen represents a low demand interactive task, e.g., video playback; and so on.

This component also considers a pre-condition given by the start battery levels (not necessarily the same) for the *m* smartphones in the set. Thus, the component also receives the given start battery level for each smartphone in the set as input data, where this level is represented as an integer in the range [0,100]%.

### 3.2. The Designed Evolutionary Algorithm

Once the component receives all the input data detailed in Section 3.1, it applies our designed evolutionary algorithm. This algorithm explores different sequences of charge/discharge actions that, at experimentation time, could be applied on the smartphones in the set to reach the given start battery levels, with the aim of finding the sequences of actions that reach these levels at the minimal possible time for the set of smartphones. In other words, this implies minimizing the preparation time for these smartphones under the given pre-conditions.

This evolutionary algorithm begins by creating an initial population with *s* feasible encoded solutions. Each solution encodes and represents sequences of charge/discharge actions that could be applied on the *m* smartphones to achieve the given start battery levels based on the current battery levels of the smartphones. Then, each solution in this population is evaluated by a fitness evaluation process, according to the considered objective: reaching the given start battery levels in the minimum possible amount of time for the set of smartphones. Then, the algorithm develops an iterative behavior until reaching the stop condition.

In each iteration, a parent selection process is applied to the current population to determine which solutions of this population will compose the mating pool, and therefore will be utilized to generate new encoded solutions. In this respect, the well-known tournament selection process [6] is applied with a tournament size *k*, to promote the selection of diverse high-fitness solutions regarding the sequences of actions indicated for the *m* smartphones. Then, the solutions in the mating pool are paired, and a crossover process is applied on each pair of solutions with a probability *P_c_* to generate a pool of new solutions. In this sense, we designed a crossover process feasible for the used encoding of solutions, which generates new solutions by interchanging the sequences of actions indicated for the *m* smartphones in the parent solutions. Then, a mutation process is applied on each new solution with a probability *P_m_* to incorporate diversity in the pool of new solutions. In this respect, we designed a mutation process feasible for the used encoding of solutions, which generates changes in the sequences of actions indicated for the *m* smartphones. After that, each new solution is evaluated by the fitness evaluation process. Then, a survival selection process is applied on the current population and the pool of new solutions to decide which solutions will compose the new population for the next iteration. In this respect, the well-known steady-state selection process is applied [6], with a replacement percentage *r*, in order to preserve the best solutions obtained by the algorithm so far.

Once the algorithm reaches the stop condition (i.e., after a given number of iterations, the algorithm outputs the best solution in the last population as result). This solution is used by the component to automatically prepare the whole set of *m* smartphones.

#### 3.2.1. Encoding of Solutions

Each solution in the population of the algorithm is represented as an *m*-tuple <*s*_1_, *s*_2_, …, *s_m_*>, where *m* is the number of smartphones considered, i.e., attached to the platform at the time the synchronized test starts. Then, the term *s_i_* (*i* = 1,…, *m*) represents a feasible sequence of charge/discharge actions (*a_i_*_1_, *a_i_*_2_, …, *a_in(i)_*) for smartphone *i*, which allows to reach *sbl*(*i*) (the start battery level corresponding to *i*) from *cbl*(*i*) (the current battery level of *i*). When *cbl*(*i*) is lower than *sbl*(*i*), *cbl*(*i*) must be increased; thus, sequence *s_i_* only includes battery charge actions. Otherwise, when *cbl*(*i*) is higher than *sbl*(*i*), *cbl*(*i*) must be decreased; thus, sequence *s_i_* only includes discharge actions. In both cases, the number, *n(i)*, of actions of the sequence *s_i_* is calculated as detailed in Equations (1) and (2).
(1)$$n\left(i\right)=actions\left(i\right)$$
(2)$$actions\left(i\right)=\left|\;cbl\left(i\right)-sbl\left(i\right)\right|$$

Each action *a_ij_* (*j* = 1,…,*n(i)*) is represented as a tuple with five elements: <*k_action*, *initial_level*, *finish_level*, *CPU_load*, *screen_state*>, where *k_action* refers to the kind of battery-related action (i.e., charge/discharge), *initial_level* indicates the initial battery level for the action, and *finish_level* indicates the end battery level for the action. The element *CPU_load* refers to the CPU load under which the action increases/decreases from/to *initial_level*. As mentioned in Section 3.1, CPU load belongs to the set {0, 30, 50, 75, 100}%. Finally, the element *screen_state* refers to the screen state (i.e., on/off) under which the action increases/decreases from/to *initial_level*.

Thus, the length of an encoded solution <*s*_1_, *s*_2_, …, *s_m_*> is calculated as detailed in Equation (3).
(3)$$length\left(\langle s_{1},s_{2},\ldots ,s_{m}\rangle \right)=\sum_{i=1}^{m}n\left(i\right)*5$$

Figure 5 shows a sample encoded solution where two smartphones are considered, and a test plan requiring these smartphones to start with 23% and 71% of battery, respectively. In relation to smartphone 1, it has a current battery level of 20%. Thus, the solution proposes a sequence of three charge actions for this smartphone. In this sequence, the first action increases the battery level from 20% to 21% under a CPU load of 30% with the screen on. Then, the second action increases the battery level from 21% to 22% under a CPU load of 75% with the screen off. Finally, the third action increases the battery level from 22% to 23% under a CPU load of 100% with the screen on. Thus, this sequence starts from the battery level 20% and reaches the required battery level of 23%. Regarding smartphone 2, it has a current battery level of 75%. Here, the solution proposes a sequence of four discharge actions for this smartphone. This sequence starts from the battery level 75% and reaches the required battery level of 71%.

#### 3.2.2. Fitness Evaluation Process

This process is applied to evaluate each encoded solution of the population according to the objective considered. Recall that the objective is to reach the pre-defined start battery levels for the *m* smartphones in the minimum amount of time for the set of smartphones. Considering that, as detailed below, the minimal possible time to prepare the set of *m* smartphones can be estimated as per the existence of (dis)charge profiles, then the mentioned objective means minimizing the difference (i.e., the error) between the time required by each smartphone to achieve the corresponding start battery level and the minimal possible time to prepare the set of *m* smartphones. In order to evaluate each encoded solution regarding this objective, the process follows the steps described below.

Given an encoded solution <*s*_1_, *s*_2_, …, *s_m_*>, where *s_i_* represents a feasible sequence of charge/discharge actions to prepare the smartphone *i*, the process calculates the average difference (i.e., average error) between the time of each sequence *s_i_* and the minimum amount of time to prepare the *m* smartphones. This average difference is determined by a well-known metric named mean absolute percentage error (MAPE). This metric is calculated as detailed in Equation (4), where *t*(*s_i_*) refers to the time (milliseconds) of *s_i_* and *T* is the minimal possible time (milliseconds) to prepare the *m* smartphones. The term *t*(*s_i_*) is calculated by Equation (5), where *t*(*a_ij_*) represents the time (milliseconds) required to develop the action *a_ij_*. In this respect, the times of actions *a_ij_* are provided by the charge/discharge profiles of smartphone *i*.

The term *T* is calculated by Equation (6), where *min*(*i*) refers to the minimum amount of time in milliseconds (i.e., optimal time) for preparing smartphone *i*. Then, the maximum of these minimum preparation times defines the minimal possible time to prepare the whole set of *m* smartphones so they all reach the next battery level as specified in the test plan being exercised. Regarding min(*i*), this time is provided by the existing charge/discharge profiles of *i*. Specifically, when the current battery level of *i* must be charged to reach the start battery level pre-defined for *i*, the minimal possible time is achieved by charging the battery under a CPU load of 0% with the screen off. Thus, the minimal possible time to prepare *i* is provided by the charge profile inherent to a CPU load of 0% with the screen off. This is the case when the current test specifies a higher battery level than the current battery level of smartphone *i*; thus, a CPU load of 0% with the screen off means *charging* at a high rate. On the other hand, when the current battery level of *i* must be discharged to reach the start battery level pre-defined for *i*, the minimal possible time is achieved by discharging the battery under a CPU load of 100% with the screen on. Therefore, the minimal possible time to prepare *i* is provided by the discharge profile inherent to a CPU load of 100% with the screen on. This is the case when the current test specifies a lower battery level than the current battery level of smartphone *i*; thus, a CPU load of 100% with the screen on means *discharging* at a high rate.

By applying Equation (4), an MAPE value higher than or equal to 0% is assigned as the fitness value of each encoded solution. Better MAPE values (i.e., lower MAPE values) are assigned to those solutions in which the times, *t*(*s_i_*), of the sequences *s_i_* detailed for preparing the *m* smartphones are closest to *T*. In these solutions, the times, *t*(*s_i_*), of the sequences *s_i_* are also closer to each other. Therefore, these solutions minimize the discharge of the *m* smartphones once these are prepared according to the sequences *s_i_* (i.e., once the start battery levels, *sbl*(*i*), are reached).

Figure 6 shows the MAPE values and the times *t*(*s_i_*) of the sequences *s_i_* of two feasible solutions A and B for an example case where a set of four smartphones have to be prepared. These solutions differ considerably regarding their MAPE value. In this sense, the MAPE value of solution A (0.01%) is much better (i.e., much lower) than that of solution B (52.37%). This is because the times, *t*(*s_i_*), of solution A (i.e., *t*(*s*_1_) = 16,575,027, *t*(*s*_2_) = 16,575,096, *t*(*s*_3_) = 16,584,583, and *t*(*s*_4_) = 16,575,097) are closer to *T* than those of solution B (i.e., *t*(*s*_1_) = 5,819,963, *t*(*s*_2_) = 16,575,096, *t*(*s*_3_) = 1,623,840, and *t*(*s*_4_) = 7,560,004). As a result, the times, *t*(*s_i_*), of solution A are closer to each other. Specifically, the times, *t*(*s_i_*), of solution A have very low differences among them (i.e., differences are in the range of [1,9556] ms), whereas the times, *t*(*s_i_*), of solution B have very significant differences among them (i.e., the differences are in the range of [9,015,092, 14,951,256] ms).

This means than these solutions differ with respect to the discharge of the smartphones once the start battery levels, *sbl*(*i*), are reached. In this sense, in solution A, the time *t*(*s_3_*) is the maximum of the times *t*(*s_i_*). Therefore, smartphone 1 will reach *sbl*(1) at *t*(*s*_1_), and so it will be prepared before smartphone 3 reaches *sbl*(3). Then, given that the time between *t*(*s*_1_) and *t*(*s*_3_) (i.e., 9556 ms) is not enough to decrease *sbl*(1), smartphone 1 will maintain *sbl*(1) until smartphone 3 reaches *sbl*(3). In a similar way, smartphone 2 will maintain *sbl*(2) and smartphone 4 will maintain *sbl*(4) until smartphone 3 reaches *sbl*(3). Thus, solution A prevents the discharge of the start battery levels *sbl*(*i*) reached by the smartphones. Unlike solution A, in solution B, the time *t*(*s*_2_) is the maximum of the times *t*(*s_i_*). Thus, smartphone 4 will reach *sbl*(4) at *t*(*s*_4_), and it will be prepared before smartphone 2 reaches *sbl*(2). Then, since the time between *t*(*s*_4_) and *t*(*s*_2_) (i.e., 9,015,092 ms) is long enough to decrease *sbl*(4), smartphone 4 will not be able to maintain *sbl*(4) until smartphone 2 reaches *sbl*(2). Likewise, smartphone 1 will not be able to maintain *sbl*(1) and smartphone 3 will not be able to maintain *sbl*(3) until smartphone 2 reaches *sbl*(2). Note that the decrease in the levels of *sbl*(4)/*sbl*(1)/*sbl*(3) (in battery units) depends on the time between *t*(*s*_4_)/*t*(*s*_1_)/*t*(*s*_3_) and *t*(*s*_2_). The longer the time between *t*(*s*_4_)/*t*(*s*_1_)/*t*(*s*_3_) and *t*(*s*_2_), the higher the decrease in *sbl*(4)/*sbl*(1)/*sbl*(3). Therefore, solution B enables the discharge of the start battery levels *sbl*(*i*) reached by the smartphones. All in all, solution A outperforms solution B in terms of minimizing the discharge of the smartphones once the start battery levels *sbl*(*i*) are reached.
(4)$$MAPE\left(\langle s_{1},s_{2},\ldots ,s_{m}\rangle \right)=\frac{100}{m}\sum_{i=1}^{m}\frac{\left|T-t\left(s_{i}\right)\right|}{T}$$
(5)$$t\left(s_{i}\right)=\sum_{j=1}^{n(i)}t\left(a_{ij}\right)$$
(6)$$T=\max_{i=1}^{m}\;\left(\min\left(i\right)\right)$$

#### 3.2.3. Crossover Process

A crossover process is applied to generate new encoded solutions from the solutions in the mating pool. In this respect, the evolutionary algorithm decides which solutions from the current population will compose the mating pool by applying the well-known tournament selection process [6]. Then, the solutions in the mating pool are organized in pairs, and the crossover process is applied on each pair of solutions with a probability *P_c_* to generate new encoded solutions. We designed a crossover process feasible for the encoding of solutions presented in Section 3.2.1. The details of this process is described below.

Given two encoded solutions, *p*1 and *p*2, the crossover process generates two new encoded solutions, *o*1 and *o*2, by following the next iterative behavior. For each one of the smartphones *i*, this process considers the sequences of actions *s_i_* detailed in *p*1 and *p*2 for *i*, and after that analyzes one by one the actions, *a_ij_*, of each sequence. For each action, *a_ij_*, of the sequence *s_i_* detailed in *p*1(*p*2), the process considers the five elements which compose the action and then copies the values detailed in *p*1(*p*2) for *k_action*, *initial_level*, and *finish_level* to the new solution *o*1(*o*2), in the same positions for these values in *p*1(*p*2). Regarding the values detailed in *p*1(*p*2) for the elements *CPU_load* and *screen_state*, each of these values is copied to the new solution, *o*1(*o*2), according to a given probability, *u*. Specifically, for each of the mentioned elements, the process generates a random number in the range of [0,1]. If this number is lower than *u*, the process copies the value detailed in *p*1(*p*2) for the element to *o*1(*o*2), in the same position for this value in *p*1(*p*2). Otherwise, if this number is higher than or equal to *u*, the process copies the value detailed in *p*1(*p*2) for the element to *o*2(*o*1), in the same position for this value in *p*1(*p*2). Therefore, this crossover process allows the generation of two new encoded solutions by interspersing the values detailed in *p*1 and *p*2 for the elements of the actions of each sequence.

Figure 7 shows an example of the crossover process. In this example, the process is applied on the encoded solutions *p*1 and *p*2 and generates the encoded solutions *o*1 and *o*2. Bold values in *o*1 and *o*2 indicate values interchanged between *p*1 and *p*2.

#### 3.2.4. Mutation Process

A mutation process is applied to each solution obtained by the crossover process with a probability *P_m_*, so as to incorporate diversity into the pool of new encoded solutions, and thus to preserve the diversity of the population throughout the generations of the algorithm. We designed a mutation process feasible for the encoding of solutions presented in Section 3.2.1. The behavior of this process is described below.

Considering an encoded solution *p*1, our mutation process generates a new encoded solution *o*1, by following the next iterative behavior. For each of the smartphones *i*, this process considers the sequence of actions, *s_i_*_,_ detailed in *p*1 for *i*, and then analyzes one by one the actions, *a_ij_*, of this sequence. For each action, *a_ij_*, the process considers the five elements that compose the action. Then, the process copies the values detailed in *p*1 for *k_action*, *initial_level*, and *finish_level* to the new solution *o*1, in the same positions for these values in *p*1. In relation to the values detailed in *p*1 for the elements *CPU_load* and *screen_state*, each one of these values is copied to the new solution *o*1 according to the mutation probability, *P_m_*. In particular, for each one of the two mentioned elements, the process generates a random number in the range of [0,1]. When this number is higher than *P_m_*, the process copies the value detailed in *p*1 for the element to *o*1 to the same position for this value in *p*1. On the other hand, when this number is lower than or equal to *P_m_*, the process does not copy the value detailed in *p*1 for the element to *o*1. In this case, the process randomly chooses other possible value for the element, and then copies this value to *o*1 to the same position for the value detailed in *p*1. Thus, this mutation process allows the generation of a new encoded solution by changing the values detailed in *p*1 for the elements of the actions of each sequence according to *P_m_*.

Figure 8 shows an example of the mutation process. In this example, the process is applied on the encoded solution *p*1, and generates the encoded solution *o*1. Bold values in *o*1 indicate new values compared to those in *p*1.

## 4. Computational Experiments

As described in Section 3, the new software component proposed for Motrol uses the described evolutionary algorithm to automatically prepare a given set of smartphones. Thus, we developed computational experiments to evaluate the performance of this evolutionary algorithm on different instances of the addressed problem. The preparation time was simulated using battery profiles as described in Section 3.1. By interleaving excerpts of different profiles of the same smartphone it is possible to realistically evaluate the effect of solutions proposed by the evolutionary algorithm.

In Section 4.1, we present the example sets utilized to develop these experiments. In Section 4.2, we detail the experimental settings defined for these experiments. Finally, in Section 4.3, we present and analyze the obtained results.

### 4.1. Instance Sets

As explained in Section 3, given a set of *m* smartphones where each smartphone *i* must (dis)charge from its current battery level, *cbl*(*i*), to a target (also called start) battery level, *sbl*(*i*), the problem consists of finding the sequences of charge/discharge actions to be applied on the *m* smartphones, in order to reach the target *sbl*(*i*) levels at the same point in time and in the minimum possible amount of time.

To evaluate the performance of the evolutionary algorithm on diverse realistic experimental scenarios, we defined 27 sets of instances of the addressed problem. Each of these instance sets contains 10 different instances. Each instance of these sets contains a number of smartphones to be prepared. The 27 defined instance sets correspond to diverse realistic preparation scenarios in terms of the number of smartphones to be prepared (aspect S) and battery level actions (aspect A and aspect V) required to prepare the smartphones. In this sense, the category of an instance is defined in relation to these three aspects that are described below.

***Aspect S***. This aspect refers to the number of smartphones *m* considered in the instance. Here, this number belongs to the following set: {4, 8, 16}. Considering that the local facet of Dew computing comprises on-premise devices within the boundaries of a wireless local area network, aspect S corresponds to potential on-premise devices in a Dew computing setting, for which we selected 16 as an upper limit representing the number of devices a domestic router can handle without compromising the delivered QoS. Thus, three different categories were considered in relation to the number of smartphones, namely S4, S8, and S16.

***Aspect A***. This aspect refers to the total number of charge/discharge actions (1 percent battery level changes) to be applied by the *m* smartphones considered in the set to reach the desired start battery levels (i.e., the sum of current-to-target battery level differences of all smartphones considered in the instance). This number is named *A*, and is calculated as detailed in Equation (7), where *actions*(*i*) refers to the number of actions to be applied by smartphone *i*. Note that *actions*(*i*) is calculated as detailed in Equation (2) (Section 3.2.1). Thus, *A* belongs to the value range [1**m*, 100**m*]. In this value range, the value 1**m* corresponds to the instances where each smartphone *i* must apply only one action to reach the desired start battery level from the current battery level, and the value 100**m* corresponds to the instances where each smartphone *i* must apply 100 actions to reach the start battery level from the current battery level. This value range has been divided into three distinct subranges in order to consider three different categories in relation to *A*. Specifically, this range has been divided into the following subranges: [1**m*, 40**m*], [41**m*, 70**m*], and [71**m*, 100**m*], in order to define the categories Low A (LA), Medium A (MA), and High A (HA) with similar subrange width, respectively.
(7)$$A=\sum_{i=1}^{m}actions\left(i\right)$$

***Aspect V***. This aspect refers to the variation in the number of charge/discharge actions among the *m* smartphones considered in the set. In other words, it is a way to differentiate problem instances based on how the total number of actions (Aspect A) is distributed among participating smartphones. That is, a smartphone set where a few smartphones have to perform the majority of the total number of actions presents a quite different challenge regarding all of them performing the same percentage of the total number of actions. This variation is measured by a well-known metric named coefficient of variation (CV). This metric is calculated as detailed in Equation (8), where *M* refers to the average number of actions to be applied by the *m* smartphones (Equation (10)) and *SD* refers to the standard deviation of the number of actions to be carried out by the *m* smartphones (Equation (9)). Thus, this metric provides a real value in the range of [0,100]%. In this range, the value 0% corresponds to the instances where the *m* smartphones must develop the same number of actions to reach *sbl*(*i*) from *cbl*(*i*) (i.e., there is no variation regarding the number of actions among the *m* smartphones). This range also has been divided into three distinct subranges in order to consider three different categories in relation to V. In particular, this range has been divided into the following subranges: [0,10]%, [10,50]%, and [50,100]%, which determines the categories Low V (LV), Medium V (MV), and High V (HV), respectively.
(8)$$CV=\frac{SD}{M}\;*\;100$$
(9)$$SD=\sqrt{\frac{\sum_{i=1}^{m}{\left(actions\left(i\right)-M\right)}^{2}}{m}}$$
(10)$$M=\frac{\sum_{i=1}^{m}actions\left(i\right)}{m}$$

Table 1 details the characteristics of the 27 defined instance sets regarding the three aspects previously mentioned. Column 1 indicates the name of each set. Column 2 details the value of the instances of each set in relation to aspect S. Columns 3 and 4 detail the value range of the instances of each set in relation to the aspects A and V, respectively. Finally, Column 5 indicates the number of instances of each set.

### 4.2. Experimental Setting

We ran the evolutionary algorithm on each one of the 10 instances of each instance set presented in Table 2. Considering that evolutionary algorithms are non-deterministic by nature, we ran this evolutionary algorithm several times (i.e., 30 runs) for each instance to obtain reliable statistical results. For each run, we recorded the solution provided by the algorithm for the instance under evaluation, the MAPE value of this solution, and the computing time taken by the algorithm to obtain this solution.

To run the evolutionary algorithm, we used the parameter settings detailed in Table 2. These parameter settings were defined based on preliminary experiments. In these experiments, we considered different parameter settings, which are usually suggested in the technical literature on evolutionary algorithms [6]. Table 3 details the parameter settings considered. Then, for each of these parameter settings, we ran the evolutionary algorithm several times (i.e., 30 runs) for each instance and we calculated the average MAPE value of the 30 solutions obtained for each instance. These experiments showed that the parameter setting detailed in Table 2 yielded the best average MAPE values for the evaluated instances.

In relation to the parameter *P_m_*, note that one of the considered settings is 1/L. The term L refers to the number of elements on which the mutation is applied, with *P_m_*, in an encoded solution. In this case, as detailed in Section 3.2.4, given an encoded solution, the mutation is applied, with *P_m_*, on two elements of each action of each one of the *m* smartphones considered in the solution. Thus, L is equal to 2*(*n*(1) + *n*(2) + … + *n*(*m*)), where *n*(*i*) is the number of actions corresponding to the smartphone *i* in the encoded solution, as described in Section 3.2.1.

### 4.3. Current Method to Prepare Smartphones in the Context of Motrol

To contextualize the performance of the evolutionary algorithm, we considered the method currently used by Motrol to determine the preparation of a given set of *m* smartphones for synchronized test plans. For simplicity, we will refer to this method as the Motrol method.

Given a set of *m* smartphones, this method defines a feasible solution to prepare the *m* smartphones, considering the current battery level, *cbl*(*i*), of each smartphone *i* in the set and the start battery level, *sbl*(*i*), to be reached by each smartphone *i* in the set. Specifically, for each smartphone *i*, this method defines one charge/discharge action, *a_i_*, which allows the battery to reach the level *sbl*(*i*) from *cbl*(*i*) in the minimal possible amount of time for the smartphone *i*. When *cbl*(*i*) is lower than *sbl*(*i*), *cbl*(*i*) must be increased to reach *sbl*(*i*). Thus, the method defines a charge action, *a_i_*, which must be developed from the level *cbl*(*i*) and must continue until *sbl*(*i*) is reached. In addition, *a_i_* must be developed under a CPU load of 0% with the screen off to achieve *sbl*(*i*) in the minimum possible time for *i*. Otherwise, when *cbl*(*i*) is higher than *sbl*(*i*), *cbl*(*i*) must be decreased to reach *sbl*(*i*). Therefore, the method defines a discharge action, *a_i_*, which must be carried out from *cbl*(*i*) and must continue until *sbl*(*i*) is reached. Moreover, *a_i_* must be carried out under a CPU load of 100% with the screen on in order to achieve *sbl*(*i*) in the minimum possible amount of time for *i*. Finally, this method provides a solution, which is composed of the *m* actions *a_i_* defined to prepare the *m* smartphones.

We applied this method on each one of the ten instances of each instance set. For each instance, we recorded the solution obtained by this method and we calculated the MAPE value of this solution in order to compare this value with those of the 30 solutions obtained by the evolutionary algorithm for each instance.

It is worth mentioning that this method differs from the proposed evolutionary algorithm in two main aspects. First, the method defines one action, *a_i_*, for each smartphone *i*, which allows Motrol to reach *sbl*(*i*) from *cbl*(*i*) in the minimum possible amount of time for the smartphone *i*. Thus, the action, *a_i_*, of each smartphone *i* is defined independently of the actions of the other smartphones in the set. Unlike this, the evolutionary algorithm explores many different sequences, *s_i_*, of actions for each smartphone *i*, with the aim of finding a sequence, *s_i_*, to collectively reach *sbl*(*i*) from *cbl*(*i*) at time *T*. As described in Section 3.2.2, *T* is the minimum possible time-taken to prepare the set of *m* smartphones. Second, this method does not consider the difference among the times of the actions, *a_i_*, defined for the *m* smartphones. In contrast to this, the evolutionary algorithm considers the difference between the time of the sequence *s_i_* determined for each of the *m* smartphones and the time *T*, and consequently considers the difference among the times of the sequences *s_i_*, which is important to minimize the discharge of the *m* smartphones once these are prepared according to the sequences *s_i_* (i.e., once the start battery levels *sbl*(*i*) are reached).

### 4.4. Experimental Results

In Table 4, we present the main results obtained from the conducted computational experiments. Column 1 indicates the name of each instance set used in the experiments. columns 2 and 3 detail the average MAPE value of the solutions obtained by the evolutionary algorithm and the Motrol method for the instances of each set, respectively. Then, columns 4 and 5 detail the maximum MAPE value obtained by the evolutionary algorithm and the Motrol method for each instance set, respectively. Finally, columns 6 and 7 detail the minimum MAPE value obtained by the evolutionary algorithm and the Motrol method for each instance set, respectively.

From Table 4, it can be seen that the average MAPE value of the solutions obtained by the evolutionary algorithm for each instance set is much lower than that of the solutions obtained by the Motrol method. For each of the instance sets S4_*A_*V, the difference between the average MAPE values achieved by the evolutionary algorithm and the Motrol method is in the range of [25.84, 35.88] %. Similarly, for each of the instance sets S8_*A_*V and S16_*A_*V, the measured difference is in the range of [27.9, 42.13] % and [29.02, 39.97] %, respectively. These results regarding the average MAPE value are mainly due to the following reasons. For most instance sets (i.e., 22 of the 27 sets), the maximum MAPE value achieved by the evolutionary algorithm is lower than the minimum MAPE value obtained by the Motrol method. In addition, for each of the 270 instances used, the MAPE values of the 30 solutions obtained by the runs of the evolutionary algorithm are significantly lower than the MAPE value of the solution provided by the Motrol method. The statistical significance of these results was ascertained by the Mann–Whitney U test, with a confidence level of α = 0.001.

Based on these results, the solutions achieved by the evolutionary algorithm decrease the MAPE value for the instances considered. This means that, according to the MAPE definition given in Section 3.2.2, these solutions reduce the difference between the preparation time of each of the *m* smartphones and the time *T*, and as a consequence, reduce the difference among the preparation times of the *m* smartphones. Therefore, as described in Section 3.2.2, these solutions will reduce the discharge of the *m* smartphones once these are prepared (i.e., once the start battery levels, *sbl*(*i*), are reached). The smaller the difference among the preparation times of the *m* smartphones, the lower the amount of unnecessary (dis)charge actions of the *m* smartphones once these are prepared.

Considering the above, in Table 5, we present the average, maximum, and minimum RPD (relative percentage difference) of the solutions achieved by the evolutionary algorithm, regarding the solutions provided by the Motrol method and in terms of the estimated discharge of the *m* smartphones once these are prepared. The metric RPD computes the average percentage difference of the estimated discharge of the *m* smartphones once these are prepared according to *s_EA_* (i.e., the solution given by the evolutionary algorithm), regarding the estimated discharge of the *m* smartphones once these are prepared according to *s_M_* (i.e., the solution given by the Motrol method). This metric is calculated according to Equation (11), where *bud*(*s_M_*, *i*) and *bud*(*s_EA_*, *i*) refer to the estimated discharge (in battery units) of smartphone *i* once it is prepared according to the solutions of *s_M_* and *s_EA_*, respectively. When the RPD value is positive, this means that *s_EA_* has achieved a saving when compared to *s_M_* in terms of the estimated discharge (in battery units) of the *m* smartphones once these are prepared.

The value of *bud*(*s*, *i*) is estimated as follows. Given a solution, *s*, represented as the *m*-tuple <*s*_1_, *s*_2_, …, *s_m_*> described in Section 3.2.1, we first calculated the preparation time *t*(*s_i_*) of smartphone *i*, by applying Equation (5) detailed in Section 3.2.1. After that, we calculated the maximal preparation time, *MPT*(*s*), of the *m* smartphones with Equation (12), which determines the preparation time of the set of *m* smartphones. As described in Section 3.2.2, when *MPT*(*s*) is higher than *t*(*s_i_*), smartphone *i* will be prepared (i.e., smartphone *i* will reach its start battery level *sbl*(*i*)) before the smartphones that require a preparation time of *MPT*(*s*). Thus, smartphone *i* will discharge its battery level, *sbl*(*i*), during the time between *t*(*s_i_*) and *MPT*(*s*). This discharge of *sbl*(*i*) (in battery units) depends on the time between *t*(*s_i_*) and *MPT*(*s*). The longer this time, the higher the discharge of *sbl*(*i*). Therefore, we calculated this time *td*(*s*, *i*), as detailed in Equation (13). Once *td*(*s*, *i*) is calculated, the estimated discharge (in battery units), *bud*(*s*, *i*), of *sbl*(*i*) during *td*(*s*, *i*) can be determined from the battery discharge profile of smartphone *i* (i.e., battery discharge profile inherent to a CPU load of 0% with the screen off).
(11)$$RPD=\frac{100}{m}\sum_{i=1}^{m}\frac{bud(s_{M},\;i)-bud(s_{EA},\;i)}{bud(s_{M},\;i)}$$
(12)$$MPT\;(s)=\;\max_{i=1}^{m}\left(t\left(s_{i}\right)\right)$$
(13)$$td\left(s,\;i\right)=MPT\left(s\right)-t\left(s_{i}\right)$$

From the results in Table 5, it follows that the solutions obtained by the evolutionary algorithm for each instance set provide a considerable average saving (average RPD) in terms of the discharge (in battery units) of the *m* smartphones once these are prepared. For the instance sets S4_*A_*V and S8_*A_*V, the average saving is in the range of [32.81, 56.25] % and [31.40, 61.98] %, respectively. For most of the instance sets S16_*A_*V (i.e., eight out of the nine sets), the average saving is in the range of [44.69, 61.05] %. In addition, the solutions obtained by the evolutionary algorithm for each instance set also provide very good minimal and maximal savings (minimal and maximal RPD). For the instance sets S4_*A_*V and S8_*A_*V, the minimal saving is in the range of [18.75, 35.42] % and [12.50, 50.00] %, respectively. For the instance sets S16_*A_*V, the minimal saving is in the range of [12.50, 50.00] %. These results regarding the minimal saving show that for each one of the 270 instances used, the 30 solutions obtained by the runs of the evolutionary algorithm provide a saving in terms of the discharge (in battery units) of the *m* smartphones once these are prepared.

In addition to the results presented in Table 4 and Table 5, in Table 6, we present the average, maximum, and minimum computing time (in seconds) required by the evolutionary algorithm for each of the instance sets. In this sense, all the computational experiments were executed on a PC equipped with an Intel core i7-3610QM 2.3GHz CPU, a 6.00 GB memory, a 1 TB HD, and a 64-bit Windows 10 operating system. In addition, the evolutionary algorithm was implemented in Java 1.8.

As shown in Table 6, the higher the total number of actions to be applied by the *m* smartphones considered in the instance (see values of A in Table 1), the higher the computing time (in seconds) required by the evolutionary algorithm. This is mainly because of the following reason. As described in Section 3.2, the main processes of the evolutionary algorithm (i.e., fitness evaluation, crossover, and mutation processes) are applied in solutions encoded as an *m*-tuple <*s*_1_, *s*_2_, …, *s_m_*>, where *s_i_* represents a feasible sequence of actions for smartphone *i*. Therefore, the computing time of these processes, and consequently the computing time of the algorithm, depend on the length of the encoded solutions. As detailed in Section 3.2.1, the length of the encoded solutions is proportional to the total number of actions to be applied by the *m* smartphones in the set. Thus, the higher the total number of actions, the higher the length of the encoded solutions; therefore, the higher the computing time of the evolutionary algorithm.

## 5. Related Work

Consumer electronic devices such as smartphones play a vital role as computing resource providers in Dew computing. Testbeds and experimentation platforms especially designed for Dew computing research are scarce. For instance, testbeds—also known as device farms—such as Firebase Test Lab, Samsung Remote Test Lab, AWS Device Farms, Sauce Labs, and Xamarin Test Cloud have been designed for developers to test mobile applications on a diverse fleet of devices. The main objective was to help developers to code their applications to support device heterogeneity. Device heterogeneity is a key aspect present in Dew computing but not the only one to be considered [17]. Other crucial aspects, particularly related to service provisioning and load balancing using clusters of consumer electronic devices, are mobility, energy management of battery and non-battery driven devices, local node collaboration, and coordination with nodes located in the fog and cloud layers. Due to the complexity of including all these aspects in a testbed, simulation is widely accepted and practiced as a performance evaluation methodology [16,17,18,19]. Simulation allows researchers to model complex interactions among entities that could be hard to realize, not to mention reproduce, with real testbeds. Particularly, Markus et al. [17] analyzed several simulation toolkits including iFogSim, MobFogSim, IoTSim-Edge, EdgeCloudSim, DewSim, and DISSECT-CF-Fog and classified them based on the abstractions and functionalities offered in relation to relevant Dew computing aspects.

Although testbeds for Dew computing are difficult to set up, these are necessary to validate simulation results. At the time of writing this paper, platforms for automating Dew computing experiments using testbeds are in the very early development stage. However, advances in other related research fields can be applied towards building a platform that holistically provides the means for experimenting and validating results in Dew computing research. One of these advances is in the line of achieving reproducibility of device mobility, a problem originally tackled in MANETs research field via the application of mobile robot technology [20]. Others refer to interfacing nodes communicating using different wireless protocols through web services to facilitate the evaluation of local collaboration among sensor and computing nodes, and the efficiency of message dissemination protocols. Concerns such as these are relevant for wireless sensor networks and IoT research; CaBIUs [21] and Indriya2 [22] are examples of recent testbeds providing such facilities.

With regard to achieving reproducibility of battery-driven Dew experiments, we have initiated efforts with a software–hardware toolkit [7] that provides researchers with basic support to (dis)charge smartphones based on a configured battery level. Such functionality has been vital to automate battery profiling whose resulting traces are used as elementary input for Dew simulations. After exploratory research using simulations, results validation becomes a necessary step to advance state-of-the-art load balancing strategies for executing tasks in clusters of smartphones and to measure the impact on energy utilization of individual smartphones. However, to make a test reproducible and the effect of load balancing strategies comparable, a feature to reset all smartphones battery-related states to the values configured in the testing scenario is required. Such a reset feature hinders the necessary (dis)charging speed synchronization between all smartphones, which is currently achieved with human intervention. To advance the automation of this necessary feature, we proposed an evolutionary algorithm that prepares smartphone battery states to run live Dew computing scenarios on real testbeds.

## 6. Conclusions and Future Work

With the aim of covering a gap in the area of Dew computing associated with reproducing battery-driven tests on real testbeds, we proposed an evolutionary algorithm to tackle the problem of preparing batteries of smartphone clusters. This state-of-the-art method using Motrol allows researchers to (dis)charge smartphones to a configured battery level. However, applying such a method to a cluster of smartphones requires a fine coordination of smartphone (dis)charging speeds, which is currently executed with human intervention. This makes the process of running battery-driven tests error prone and difficult to scale and reproduce. We designed an algorithm to perform such a preparation, consequently allowing Dew researchers not only to automate the execution of a series of battery-driven tests, e.g., the evaluation of several load balancing strategies, but also to simultaneously reduce the time taken to start each test.

The performance of the proposed algorithm was evaluated by simulating battery behavior using real smartphone traces and covering 270 combinations of different battery levels and smartphone cluster sizes. The average MAPE obtained with the evolutionary algorithm was 12.84%, indicating that (dis)charging plans differ on average by around 12% of the time the slowest smartphone takes to reach the start battery level. This represents a considerable improvement over the MAPE value obtained with the default approach to prepare smartphones, called the Motrol method, which was 46.05% on average. Moreover, when comparing the amount of battery units that smartphones vary by from the time each one reaches the start battery level until all in the set are prepared, the evolutionary algorithm reports between 12 and 61% savings on average w.r.t the Motrol method, meaning that the evolutionary algorithm prepares a test with less battery variations.

In future work, we plan to design an approach with a global view of both the preparation and test execution, assuming several preparation scenarios along with test execution estimations as input. The resulting sequence might involve different scenarios, i.e., scenarios that do not necessarily share the same battery conditions. It is interesting to study to what extent an approach such as this saves time and battery changes compared to plans with a local view of scenario preparation as those derived from the proposed evolutionary algorithm. Moreover, in the experiments, the computational time of solutions provided by the evolutionary algorithm increases with the number of smartphones and the total amount of battery changes involved in the preparation scenario, reaching nearly fifty seconds for the most complex tested scenario. Therefore, an extension we are planning for this research is improving the execution time of the algorithm by parallelizing the exploration of the solution space [23].

## Figures and Tables

**Figure 1 sensors-23-01388-f001:**
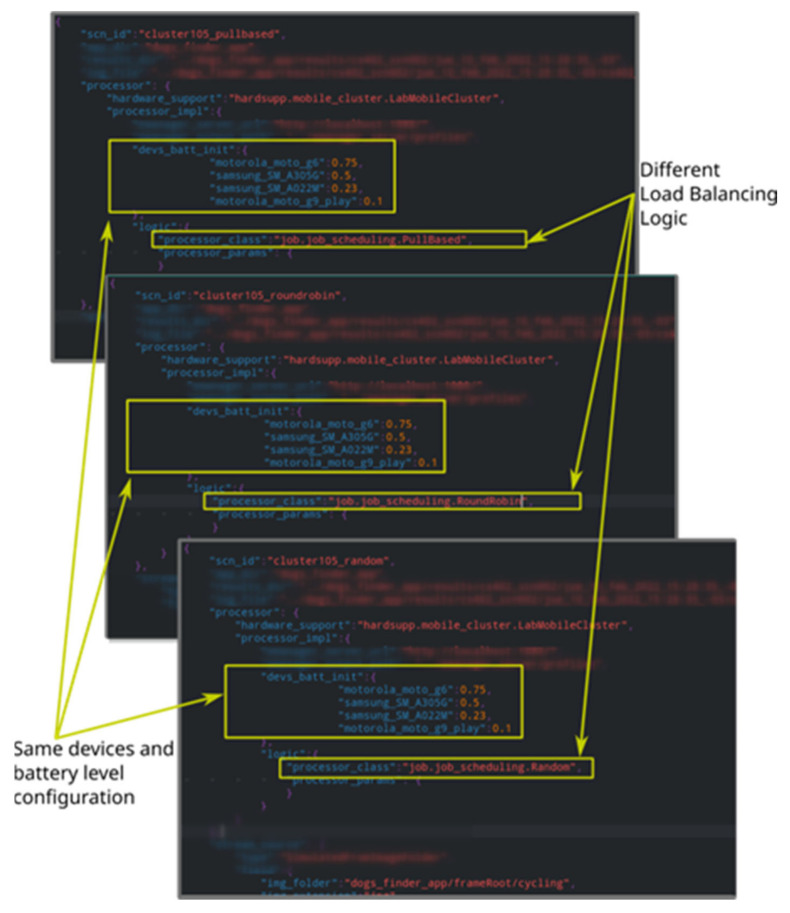
Synchronized test specification examples.

**Figure 2 sensors-23-01388-f002:**
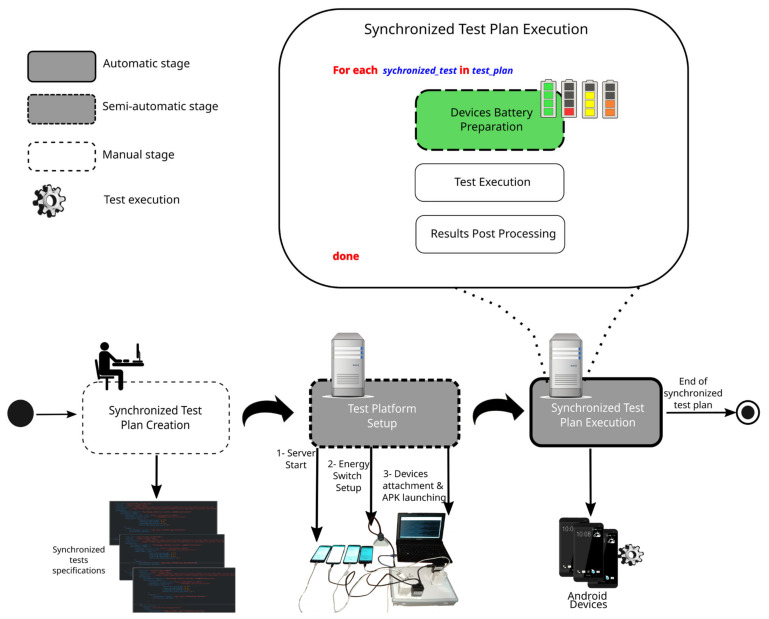
Overview of Motrol synchronized test plan execution workflow.

**Figure 3 sensors-23-01388-f003:**
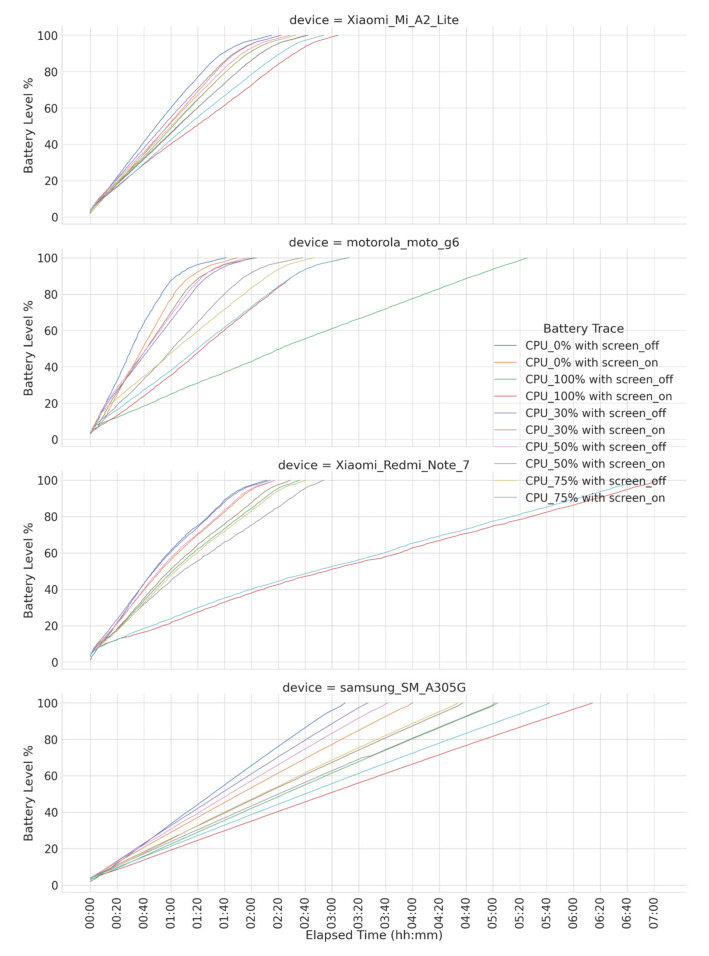
Charging traces for smartphones considered in the experiments.

**Figure 4 sensors-23-01388-f004:**
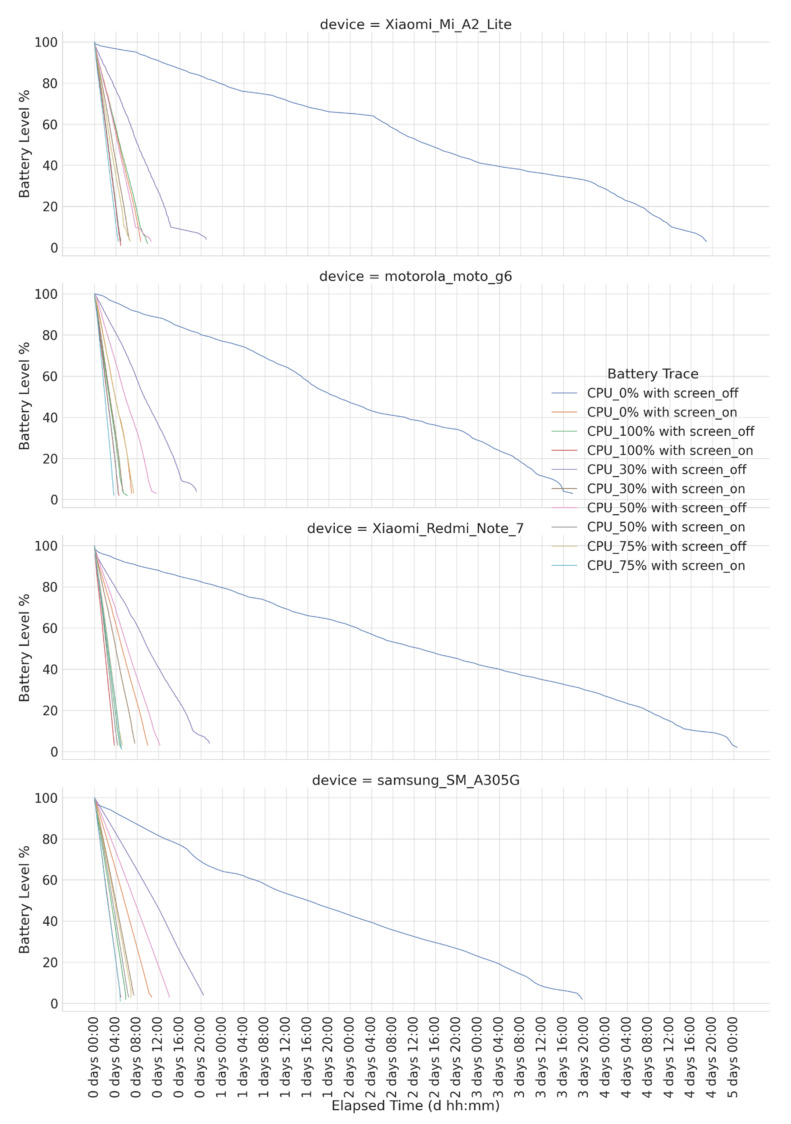
Discharging traces for smartphones considered in the experiments.

**Figure 5 sensors-23-01388-f005:**
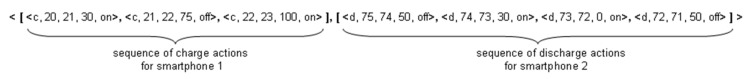
Encoded solution for an example case.

**Figure 6 sensors-23-01388-f006:**
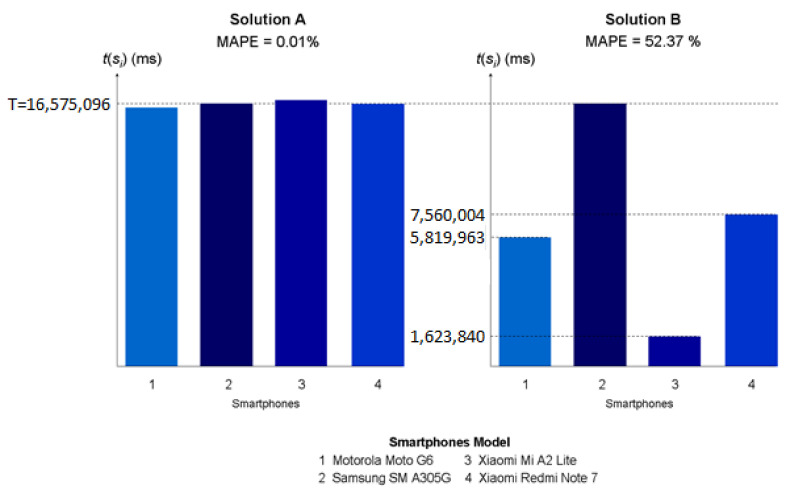
MAPE values and times *t*(*s_i_*) of two feasible solutions for an example case (Y axis is in milliseconds).

**Figure 7 sensors-23-01388-f007:**
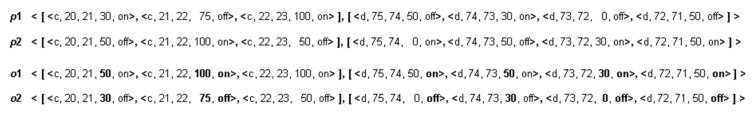
Example of the crossover process.

**Figure 8 sensors-23-01388-f008:**
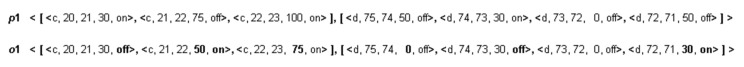
Mutation process example.

**Table 1 sensors-23-01388-t001:** Characteristics of the defined instance sets.

Instance Set	S	A	V (%)	Nr. of Instances
S4_LA_LV	4	[4, 160]	[0, 10]	10
S4_LA_MV	4	[4, 160]	(10, 50]	10
S4_LA_HV	4	[4, 160]	(50, 100]	10
S4_MA_LV	4	[164, 280]	[0, 10]	10
S4_MA_MV	4	[164, 280]	(10, 50]	10
S4_MA_HV	4	[164, 280]	(50, 100]	10
S4_HA_LV	4	[284, 400]	[0, 10]	10
S4_HA_MV	4	[284, 400]	(10, 50]	10
S4_HA_HV	4	[284, 400]	(50, 100]	10
S8_LA_LV	8	[8, 320]	[0, 10]	10
S8_LA_MV	8	[8, 320]	(10, 50]	10
S8_LA_HV	8	[8, 320]	(50, 100]	10
S8_MA_LV	8	[328, 560]	[0, 10]	10
S8_MA_MV	8	[328, 560]	(10, 50]	10
S8_MA_HV	8	[328, 560]	(50, 100]	10
S8_HA_LV	8	[568, 800]	[0, 10]	10
S8_HA_MV	8	[568, 800]	(10, 50]	10
S8_HA_HV	8	[568, 800]	(50, 100]	10
S16_LA_LV	16	[16, 640]	[0, 10]	10
S16_LA_MV	16	[16, 640]	(10, 50]	10
S16_LA_HV	16	[16, 640]	(50, 100]	10
S16_MA_LV	16	[656, 1120]	[0, 10]	10
S16_MA_MV	16	[656, 1120]	(10, 50]	10
S16_MA_HV	16	[656, 1120]	(50, 100]	10
S16_HA_LV	16	[1136, 1600]	[0, 10]	10
S16_HA_MV	16	[1136, 1600]	(10, 50]	10
S16_HA_HV	16	[1136, 1600]	(50, 100]	10

**Table 2 sensors-23-01388-t002:** Parameter settings of the evolutionary algorithm.

Parameter	Value
*Population size*	100
*k* (*tournament selection*)	10
*P_c_* (*crossover*)	1.0
*P_m_* (*mutation*)	1/L
*r* (*steady-state selection*)	50%
*Number of generations or iterations*	2000

**Table 3 sensors-23-01388-t003:** Parameter settings considered in the preliminary experiments.

Parameter	Values Considered
*Population size*	{100, 200}
*k* (*tournament selection*)	{2, 5, 10}
*P_c_* (*crossover*)	{0.7, 0.8, 0.9, 1.0}
*P_m_* (*mutation*)	{1/L} U {0.1, 0.2, 0.3}
*r* (*steady-state selection*)	{25%, 50%}
*Number of generations or iterations*	{1000, 2000, 3000, 4000, 5000}

**Table 4 sensors-23-01388-t004:** Average, maximum, and minimum MAPE (%) value obtained by the evolutionary algorithm (EA) and the Motrol method (M) for each instance set. Bold values indicate better average MAPE (%) values. The symbol * indicates that the maximum MAPE (%) value reached by EA is lower than the minimum MAPE (%) value reached by M.

	MAPE (%)					
Instance Set	Average		Maximum		Minimum	
	EA	M	EA	M	EA	M
S4_LA_LV	**17.09**	46.66	64.11	70.56	3,E-04	22.31
S4_LA_MV	**12.19**	48.07	* 33.26	66.73	0.01	34.58
S4_LA_HV	**26.30**	52.14	53.49	68.72	4.63	26.70
S4_MA_LV	**5.05**	33.53	* 12.51	43.64	2,E-05	19.13
S4_MA_MV	**8.26**	37.48	25.96	47.53	1,E-04	21.26
S4_MA_HV	**17.99**	50.37	36.87	65.05	1,E-03	34.56
S4_HA_LV	**3.80**	31.43	* 10.22	45.21	2,E-05	13.45
S4_HA_MV	**5.14**	35.93	* 19.95	48.84	1,E-04	22.49
S4_HA_HV	**10.74**	45.25	* 23.86	52.37	0.01	36.29
S8_LA_LV	**20.07**	50.23	53.35	73.70	4.19	28.71
S8_LA_MV	**11.62**	52.40	* 27.99	62.00	2.32	38.80
S8_LA_HV	**15.75**	57.88	* 24.92	71.31	4.15	41.10
S8_MA_LV	**5.76**	33.66	* 11.30	45.77	3,E-04	19.21
S8_MA_MV	**16.15**	49.84	* 30.59	58.13	1.47	36.37
S8_MA_HV	**13.81**	48.21	* 27.63	55.87	0.01	36.78
S8_HA_LV	**5.72**	40.60	* 15.69	52.64	7,E-05	32.41
S8_HA_MV	**6.39**	40.93	* 19.22	57.58	0.01	29.26
S8_HA_HV	**16.51**	49.29	* 30.29	57.37	2.34	37.24
S16_LA_LV	**6.22**	35.24	*8.59	43.22	2.50	24.87
S16_LA_MV	**18.83**	58.69	* 25.76	65.45	9.37	54.63
S16_LA_HV	**25.45**	61.99	* 46.31	75.78	14.60	51.59
S16_MA_LV	**4.42**	34.97	* 10.38	50.27	2,E-03	24.70
S16_MA_MV	**15.42**	55.39	* 23.81	64.78	7.56	48.38
S16_MA_HV	**23.41**	61.45	* 36.31	67.81	14.75	53.77
S16_HA_LV	**4.87**	35.31	* 7.75	42.60	0.14	28.25
S16_HA_MV	**12.70**	46.82	* 22.34	52.11	7.57	37.75
S16_HA_HV	**16.99**	49.82	* 24.05	57.52	11.00	38.72

**Table 5 sensors-23-01388-t005:** Average, maximum, and minimum RPD (%) values obtained by the evolutionary algorithm for each instance set.

Instance Set	RPD (%)		
	Average	Maximum	Minimum
S4_LA_LV	32.81	75.00	25.00
S4_LA_MV	45.00	75.00	25.00
S4_LA_HV	33.10	75.00	25.00
S4_MA_LV	50.00	75.00	25.00
S4_MA_MV	49.54	75.00	18.75
S4_MA_HV	50.83	75.00	35.42
S4_HA_LV	56.25	75.00	25.00
S4_HA_MV	52.50	75.00	25.00
S4_HA_HV	50.21	75.00	25.00
S8_LA_LV	31.40	62.50	12.50
S8_LA_MV	51.98	87.50	25.00
S8_LA_HV	50.17	77.08	12.50
S8_MA_LV	41.25	56.25	29.17
S8_MA_MV	50.73	68.75	19.79
S8_MA_HV	41.87	62.50	18.75
S8_HA_LV	61.98	81.25	50.00
S8_HA_MV	58.44	75.00	25.00
S8_HA_HV	52.50	71.88	37.50
S16_LA_LV	12.95	15.63	12.50
S16_LA_MV	53.93	68.75	29.69
S16_LA_HV	53.58	68.75	33.33
S16_MA_LV	44.69	65.63	31.25
S16_MA_MV	61.05	83.33	49.48
S16_MA_HV	53.31	71.46	25.63
S16_HA_LV	53.39	65.63	31.25
S16_HA_MV	58.88	67.19	50.00
S16_HA_HV	54.36	69.27	41.07

**Table 6 sensors-23-01388-t006:** Average, maximum, and minimum computing time (in seconds) required by the evolutionary algorithm for each instance set.

Instance Set	Computing Time (in Seconds)		
	Average	Maximum	Minimum
S4_LA_LV	5.00	7.20	2.00
S4_LA_MV	5.96	7.60	3.60
S4_LA_HV	5.64	7.60	3.20
S4_MA_LV	9.92	13.20	5.60
S4_MA_MV	9.64	13.20	6.80
S4_MA_HV	7.88	10.00	6.00
S4_HA_LV	13.16	16.40	10.00
S4_HA_MV	11.96	14.80	9.60
S4_HA_HV	11.60	13.60	9.60
S8_LA_LV	10.32	14.80	3.60
S8_LA_MV	11.84	14.80	8.00
S8_LA_HV	11.48	14.80	8.80
S8_MA_LV	19.80	25.60	13.60
S8_MA_MV	18.12	24.40	10.80
S8_MA_HV	15.24	21.60	10.80
S8_HA_LV	26.88	35.20	19.20
S8_HA_MV	23.92	30.00	18.80
S8_HA_HV	22.64	26.80	18.80
S16_LA_LV	20.58	22.60	18.60
S16_LA_MV	20.80	22.40	19.20
S16_LA_HV	20.36	22.80	18.00
S16_MA_LV	29.88	40.00	23.20
S16_MA_MV	33.64	36.80	29.60
S16_MA_HV	28.92	34.80	23.60
S16_HA_LV	49.04	59.60	41.60
S16_HA_MV	42.32	45.20	41.20
S16_HA_HV	41.60	42.80	40.40

## Data Availability

The software and data (i.e., problem instances) used to test the proposed solution and the implemented evolutionary algorithm can be found at https://github.com/matieber/mobilebattprep, accessed on 1 October 2022.

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
