# Peer review of "Speeding up Smartphone-Based Dew Computing: In Vivo Experiments Setup Via an Evolutionary Algorithm"

_sensors, 2023, doi:10.3390/s23031388_

Round 1

Reviewer 1 Report

1、In Eq. 8,  the  multiplication operator you used is x,  not * ?  the operator should be unified in whole paper.

2、In Eq6, the definition of T is not clear, which will effect on your important parameter MAPE,etc.

3、in many places, your english isnot fluent, please modify it carefully.

Author Response

In Eq. 8,  the  multiplication operator you used is x,  not * ?  the operator should be unified in whole paper.

  • Authors’ response: We thank the reviewer for the suggestions made. We have unified this operator in the whole paper (“*”). We have also fixed the style of Equation 6.

In Eq. 6, the definition of T is not clear, which will effect on your important parameter MAPE,etc.

  • Authors’ response: We have extended paragraph #2 on page 11 when the term T is introduced in order to make its definition clearer.

In many places, your english is not fluent, please modify it carefully.

Authors’ response: We have got our manuscript checked by a native-English speaking colleague. We also did a round of general language revision.

Reviewer 2 Report

The paper is well structured and the treatment of the topic is complete and well presented. It represents the continuation of the thematic aspirations in the development of Dew Computing state and applications. The work should be revised for minor errors and the quality of the language should be improved.

 Reference calls specifying names and numbers in the list of references are not matched. For example, in Markus, the reference number is 17 instead of 16. This is repeated in various places throughout the work.

Author Response

The paper is well structured and the treatment of the topic is complete and well presented. It represents the continuation of the thematic aspirations in the development of Dew Computing state and applications. The work should be revised for minor errors and the quality of the language should be improved.

  • Authors’ response: We thank the reviewer for the encoraging comments. We have both revised the minor errors as indicated below, and we have got our manuscript checked by a native-English speaking colleague. We have also removed double whitespaces.

Reference calls specifying names and numbers in the list of references are not matched. For example, in Markus, the reference number is 17 instead of 16. This is repeated in various places throughout the work.

  • Authors’ response: The problem was due to wrong reference numbering. We have now checked that citations in the text match the correct reference. Lastly, an incorrect citation was removed.

Reviewer 3 Report

The paper presents a methodology to run battery-driven tests on smartphone cluster testbeds. The paper shows a very good approach and resolution for the proposed contribution.

I recomend to create more detalied versions of figure 2, showing each step more detailed and not only one big picture only. One for synchronized test, other for test platform, other for test plan execution

Author Response

The paper presents a methodology to run battery-driven tests on smartphone cluster testbeds. The paper shows a very good approach and resolution for the proposed contribution.

  • Authors’ response: We thank the reviewer for the encoraging comments.

I recomend to create more detalied versions of figure 2, showing each step more detailed and not only one big picture only. One for synchronized test, other for test platform, other for test plan execution

Authors’ response: Thank you for your suggestion. Please note that since the first stage – Synchronized Test Plan Creation – mainly involves test specification in JSON format, it has not steps associated in the Figure, which means a new Figure would not be necessary. We have now made this clear in the text. With respect to the second stage –Test Platform Setup- we now refer readers to a recent publication of our own [7], where the involved steps are illustrated through a video. Finally, the steps of the last stage –Test Plan Execution- are already illustrated in Figure 2.